# Research Progress on Oxidative Stress and Its Nutritional Regulation Strategies in Pigs

**DOI:** 10.3390/ani11051384

**Published:** 2021-05-13

**Authors:** Yue Hao, Mingjie Xing, Xianhong Gu

**Affiliations:** State Key Laboratory of Animal Nutrition, Institute of Animal Sciences, Chinese Academy of Agricultural Sciences, Beijing 100193, China; haoyue@caas.cn (Y.H.); 13592548970@163.com (M.X.)

**Keywords:** oxidative stress, pigs, performance, signaling pathways, nutritional additive modulation

## Abstract

**Simple Summary:**

In the process of production, especially the modern intensive scale farming where high quality and high efficiency are pursued, pigs are subjected to a series of adverse stimuli from birth to slaughter (e.g., immunotherapy, environmental changes, uncomfortable temperature, feed contamination, improper transportation, slaughter methods, etc.). These adverse stimuli eventually translate into an imbalance in redox levels in the body, resulting in oxidative stress. The generation of oxidative stress, in turn, eventually causes damage to the pigs. To eliminate/reduce this harmful effect and counteract oxidative stress, pigs use part of the energy reserved for growth, which eventually leads to a decrease in production performance and causes unnecessary economic losses.

**Abstract:**

Oxidative stress refers to the dramatic increase in the production of free radicals in human and animal bodies or the decrease in the ability to scavenging free radicals, thus breaking the antioxidation–oxidation balance. Various factors can induce oxidative stress in pig production. Oxidative stress has an important effect on pig performance and healthy growth, and has become one of the important factors restricting pig production. Based on the overview of the generation of oxidative stress, its effects on pigs, and signal transduction pathways, this paper discussed the nutritional measures to alleviate oxidative stress in pigs, in order to provide ideas for the nutritional research of anti-oxidative stress in pigs.

## 1. Introduction

The concept of oxidative stress was first mentioned by Helmut Sies in 1985 in his book titled Oxidative Stress: A disturbance in the prooxidant-antioxidant balance favors the former [1]. The prooxidant system has subsequently been developed for more than 30 years through continuous research and refinement. In a 2018 review, oxidative stress was defined as a state of imbalance between excessive (free) oxidant radicals and insufficient degradation of these radicals by antioxidant systems as an in-house defense mechanism [2]. Free radicals in living organisms are intermediates of various biochemical reactions. These are small, unstable, and diffusible small molecules that are highly chemically active due to the presence of unpaired electrons. Biological systems possess several types of radicals, the most important of which are reactive oxygen species (ROS) and reactive nitrogen radicals (RNS) [3].

Redox homeostasis is important for the organism. Under the interaction of oxidants and antioxidants, ROS levels in organisms are in dynamic equilibrium, just like pH regulation [4]. ROS is present at or below normal physiological concentrations and has a physiological role in the cellular response to hypoxia [5]. According to the current research understanding, ROS play an important role in host defense, cell signaling, and biosynthetic processes [6]. When cells are stimulated by adverse factors in the internal and external environment, excessive ROS are produced, which, when not cleared in time, result in increased concentrations and finally exceed the normal physiological range. Abnormally high levels of ROS can cause harm to tissues and cells, such as abnormal signaling pathways, energy metabolism disorders, gene mutations, and protein structure changes [7] (Figure 1), and thus, affect the functions of cells, tissues, organs, and even systems.

Various factors can induce oxidative stress in pig production. Oxidative stress is often accompanied by other pathological factors, which has a direct negative impact on pig performance and healthy growth. Therefore, studying the mechanism of oxidative stress in animals and human intervention programs are of great significance to animal and human health. This review is planned to summarize the generation of oxidative stress, its effects on pigs, signal transduction pathways of oxidative stress, and the nutritional measures to alleviate oxidative stress in pigs. The aim is to provide alternative nutritional ideas for the research of anti-oxidative stress in pigs.

## 2. Oxidative Stress in Pigs

In pig production, many factors can induce the body to produce a large number of free radicals, which can cause oxidative damage in pigs (especially piglets). Swine intestinal tract is a major target organ of free radical attack, which leads to intestinal structure destruction, microbial disturbance, and nutrient absorption obstacles, and ultimately leads to decreased feed intake and slowed down or negative weight gain of pigs, which has a serious impact on the economic benefits of pig breeding. At present, domestic and foreign studies have confirmed that there are five main factors in the pig production process that induce oxidative stress and affect the healthy growth of pigs. These factors include birth, weaning stress, mycotoxin pollution in feed, feeding environment, social factors, etc.

### 2.1. Oxidative Stress at Birth

The main changes during the sow parturition are from the fetal placenta mediated passive respiration in the hypoxic environment of the sow uterus to the spontaneous respiration in the hyperxic environment outside the uterus after parturition. At the same time, the birth process also involves changes in ambient temperature, humidity, lighting, and noise. The sudden change in these factors may cause the mitochondrial respiratory system and other physiological metabolic system of newborn piglets to produce large amounts of oxygen free radicals. However, the antioxidant system of newborn piglets is very weak, which cannot remove excessive free radicals in time, resulting in an oxidative stress reaction of newborn piglets [8]. Yin et al. (2013) studied the changes of blood oxidation indexes and antioxidant indexes of newborn piglets during 21 days after birth [9]. It was found that the level of blood lipid oxidation product malondialdehyde (MDA) was as high as 900 pmol/mg on birth, but decreased to about 200 pmol/mg on day 7 after birth. At the same time, protein and nucleic acid oxidation products also showed a peak on birth and significantly decreased on day 7. Moreover, the activities of antioxidant enzymes including glutathione peroxidase (GSH-Px) and superoxide dismutase (SOD) in piglets were low after birth, but significantly increased after 7 days. These results further confirm that large amounts of free radicals are produced during the birth process of piglets, and the weak antioxidant system cannot timely clean up the free radicals induced by birth stress, thus causing the oxidative stress response of piglets.

### 2.2. Oxidative Stress during Weaning

Previous study found that weaning stress in piglets was closely related to oxidative stress. Blood malondialdehyde (MDA) increased significantly on the third day after weaning, and protein hydroxyl, a marker of protein oxidative damage, significantly increased on the first day after weaning. These results indicate that weaning induces the oxidative stress response in piglets, the sensitivity of lipid and protein is different in the process of weaning oxidative damage, and protein is more susceptible to the effects of weaning oxidative stress [10]. Luo et al. (2016) further confirmed that the content of oxygen free radical hydrogen peroxide (H_2_O_2_) in liver significantly increased after weaning, while the activities of antioxidant enzymes such as GSH-Px and SOD were significantly inhibited [11]. Weaning oxidative stress is affected by many factors, and its mechanism is a multi-factor and multi-level complex process. Current studies have confirmed that multiple signaling mechanisms are involved in weaning oxidative stress response, including nuclear factor erythroid 2-related factor 2 (Nrf2) and mitogen-activated protein kinase (MAPK). Moreover, autophagy, nuclear factor kappa B (NF-κB) and intestinal microorganisms have been reported to be involved in weaning stress in piglets. However, its role in weaning-induced oxidative damage and its mechanism still need to be further studied.

### 2.3. Oxidative Stress Induced by Mycotoxins

A large number of studies have shown that ingestion of feed contaminated with mycotoxin can induce systemic or tissue oxidative stress in pigs [12,13,14]. Treatment of intestinal epithelial cells with deoxynivalenol (DON) can directly induce the production of oxygen free radicals, and autophagy may be the underlying mechanism. In a previous study, CRISPR-Cas9 gene editing technology was used to knock out key genes of autophagy, and the results showed that the lack of autophagy affected the expression of stress-related genes in cells [15]. Yin et al. (2015) established a H_2_O_2_-induced acute oxidative stress model in piglets and further found that oxidative stress could induce autophagy and is linked to the signaling pathways of the inhibitor of nuclear factor kappa-B kinase (IKK). It mediates the expression of antioxidant genes (such as SOD and GSH-Px) in the body [16].

Feeding DON-contaminated feed induced oxidative stress in piglets and caused a disturbance of the nutrient metabolism. However, the addition of functional amino acids (such as glutamate and arginine) promoted the body’s antioxidant capacity and alleviated oxidative stress injury [17,18,19]. Effects of mixed mycotoxins (including aflatoxin B1, DON, ochratoxin, and fumatoxin) on piglets were studied by natural fermentation and mildewing. The results showed that the mixed mycotoxin significantly reduced the performance and blood SOD activity, and addition of arginine significantly increased blood glutathione (GSH) content [20].

### 2.4. Oxidative Stress Caused by Environmental and Social Factors

Many environmental and social factors during pig production can also induce oxidative stress in pigs, including feeding density, fighting, pig house hygiene, heat and cold stress, transportation stress, and E. coli infection. In the actual production process, in order to save the feeding space, the feeding density of pigs is often greater than the standard of piggery construction. However, high-density breeding often induces a series of adverse health factors, including lack of space for activities, rising house temperature, weakened ventilation effect, accumulation of harmful gases, large number of bacteria, food and water shortage, too much fighting and biting behavior, etc. Studies have shown that high-density feeding can significantly induce the production of free radicals in pigs at different stages, thus destroying the antioxidation–oxidation balance and causing oxidative damage [21]. By comparing markers of blood oxidative damage, it was found that the blood protein hydroxyl group of high-density pigs was significantly increased, suggesting that high-density feeding environment could cause an oxidative stress response and oxidative damage of pigs [22]. There is no evidence that high-density feeding directly causes oxidative stress in pigs. However, high-density feeding induces house temperature rise, harmful gas accumulation, bacterial infection, and fighting, which can directly cause large amounts of free radicals in pigs and lead to oxidative stress injury.

Pigs are very sensitive to heat stress, so high temperature is the most common source of stress during pig production. Short-term heat stress tests (2 days) showed significant increases in rectal temperature, respiratory metabolic rate, and intestinal permeability in growing pigs, accompanied by oxidative stress responses [23]. Long-term heat stress tests (3 weeks) identified 37 liver differential proteins, 10 of which are involved in the oxidative stress response of pigs [24]. Porcine cell culture tests further confirmed that heat stress induced oxidative stress in porcine muscle cells, but did not change the inflammatory signals [25].

In addition, cold stress, transport stress, and infection can induce strong stress response and affect the metabolism of pigs. However, there are few studies on the effects of these stressors on the oxidative system of the body. Therefore, further studies on the effects of these stressors on the production of free radicals and the antioxidant system of the body are still needed.

## 3. Model of Oxidative Stress in Pigs

Research on pig oxidative stress models mainly includes the diquat model, hydrogen peroxide model, lipopolysaccharide model, vomiting toxin model, etc. Different models have different mechanisms, but they can all induce pigs to produce a large number of free radicals, thereby destroying the body’s antioxidation–oxidation balance and causing oxidative stress.

At present, the main way to establish oxidative stress in pigs is the diquat model, by intraperitoneal injection of a certain dose of diquat (8–15 mg/kg). It can not only decompose superoxide anion free radicals by itself, but also stimulate the generation of free radicals by acting on the cellular respiratory cycle, causing the disorder of the antioxidation–oxidation balance in the body environment and inducing the oxidative stress reaction in animals [26,27]. Xu et al. (2008) evaluated the oxidative stress effect of diquat injection in growing pigs, and found that diquat injection significantly induced oxidative stress response in the pig systems 7 days after diquat injection, and the stress effect lasted for 28 days [28]. The study by Cao et al. (2018) showed that diquat-induced oxidative stress increases intestinal permeability, impairs mitochondrial function, and triggers mitophagy in piglets. At present, the diquat model has been widely used to study the effects of nutritional intervention on the oxidative stress response in pigs [29].

Through gastric intubation, Yin et al. (2015) established an acute model of H_2_O_2_ oxidative stress in piglets [16]. Injection of 5–10% H_2_O_2_ significantly induced systemic oxidative stress in piglets within 2 days, and at the same time, caused intestinal damage by destroying the intestinal permeability and intestinal villus structure [16]. Although this model is directly induced by H_2_O_2_, due to the need to perform surgery on pigs in advance and the fact that H_2_O_2_ can easily cause gastrointestinal ulcer in pigs, it has great limitations in application. However, in cell experiments, H_2_O_2_ is the most important and direct inducer of oxidative stress, which is of great significance for the study of oxidative stress mechanisms in cell models [30]. It was found that H_2_O_2_ treatment could induce autophagy, mitochondrial disorders, and apoptosis in piglets or a porcine intestinal epithelial cell model, and NF-κB and Nrf2 signaling pathways were involved in H_2_O_2_-induced oxidative stress [16,31].

The lipopolysaccharide model has been widely used as an inflammatory response model in piglet studies [32,33,34], but the inflammatory response is closely related to oxidative stress. The production of free radicals in the process of inflammatory reaction is also stimulated, and oxidative stress reaction is also involved in the process of the inflammatory response [35,36]. Tang et al. (2018) used LPS to establish a cell oxidative injury model in IPEC-J2 cells [37].

Furthermore, a large number of in vivo and in vitro experiments have found that DON can induce the generation of free radicals in addition to mycotoxins, so some studies have also used DON to induce oxidative stress [15,38]. The results of Marin et al. (2018) indicated that ochratoxin A (OTA) and aristolochic acid (AA) could induce inflammation and oxidative stress in the liver and kidney of weanling piglets [39].

In addition, Li et al. (2020) established in a chronic oxidative stress pig model induced by D-galactose [40]. The results demonstrated that administration of D-gal significantly affected the growth performance and SOD and GSH-Px levels, including related mRNA expression suppression, MDA levels enhancement, gut microbiota dysfunction, and serum amino acid alteration in pigs.

## 4. Oxidative Stress Signal Pathways

Oxidative stress can disrupt intracellular reduction-oxidation (redox) levels, which in turn inhibits/activates several signaling molecules and signaling pathways, such as the Kelch-like ECH-associated protein 1 (Keap1)/Nrf2 signaling pathway [41,42,43,44], the NF-κB signaling pathway (a key regulator of protein synthesis) [45,46], brain-derived neurotrophic factor/tropomyosin-related kinase receptor type B signaling channel [47], the phosphoinositide 3-kinase (PI3K)/Akt signaling pathway [48], protein kinase C [49,50], MAPKs [51,52], adenosine 5‘-monophosphate (AMP)-activated protein kinase (AMPK), etc. These signaling molecules ultimately regulate the expression of relevant redox-related target genes to regulate redox levels. In this review, we discuss the effects of the Keap1/Nrf2, MAPKs, and AMPK signaling pathways on oxidative stress.

### 4.1. Keap1/Nrf2/ARE Signaling Pathway

#### 4.1.1. Structural Domains of Keap1 and Nrf2

Nrf2 is a member of the CNC (cap’-n’-collar) family of transcription factors with six highly conserved regions, and these homologous structures are named Neh1-Neh6 (Nrf2-ECH homology 1 to 6). The first conserved region is Neh1 that contains the CNC homologous region and the leucine zipper structure. The second and third regions are highly conserved protein amino- and carboxy-termini called Neh2 and Neh3, respectively. Furthermore, there are two conserved acidic regions (Neh4 and Neh5) and a conserved serine-rich region (Neh6) [53] (Figure 2). The Neh1 region contains a leucine zipper-like structure with the small protein molecule Maf (including Maf F, Maf G, and Maf K) to form a heterodimer that recognizes and binds to a specific base sequence (TGCTGA(G/C)TCAGCA) present on the antioxidant response element (ARE) to initiate transcription of the target gene [54]. Nrf2 deficiency induces oxidative stress [55] and it essentially protects cells from damage by upregulating the expression of cytoprotective enzymes with ARE in a series of its promoter regions [56].

Keap1 belongs to the BTB-Kelch protein family and consists of about 50 members named Kelch-like 1–42 or Kelch and BTB domain-containing 1–14 [57]. Keap1 is a cytoplasmic protein of 624 amino acids and five distinct regions, the amino-terminal region, the BTB/POZ (Bric-a-brac, tramtrac, broad-complex/poxvirus zinc finger) domain, a cysteine-rich intervening region, the double-glycine repeat or Kelch domain, and the carboxy-terminal region [58] (Figure 2). Keap1 binds to the N-terminal Neh2 domain of Nrf2 through common DLG and ETGE motifs. Upon oxidative stress, the DLG motif in Nrf2 is released from the DRG region of Keap1; thus, it blocks the ubiquitination and degradation of Nrf2 [59].

#### 4.1.2. Signaling Pathway in Keap1/Nrf2/ARE Oxidative Stress State

Nrf2 is located in the cytoplasm and Keap1 is a pro-electron oxidative stress sensor that acts as a stress receptor and a splice component of the Cullin3 (Clu3)-based ubiquitin E3 ligase. Nrf2 and Keap1 bind to each other in a 1:2 ratio, in which two Keap1 proteins form a homodimer and bind through their BTB structural domains to the Nrf2 protein [60]. Under normal (stress-free) conditions, Keap1 binds to ubiquitinated Nrf2, which leads to rapid degradation of Nrf2 via the proteasome pathway, thus inhibiting the transcriptional activity of Nrf2 and maintaining it in a low and inactive steady state in the cell [61]. Upon cellular exposure to oxidative or electrophilic stress, electrophilic reagents modify the reactive cysteine residues of Keap1, causing a loss of its ability to ubiquitinate Nrf2 and its activity as an E3 ligase component. Nrf2 is stabilized and translocated to the nucleus, where it heterodimerizes with small Maf via the antioxidant/electrophile response element (ARE/EpRE), and activates protective cellular target genes for transcription [62]. These cytoprotective genes include: (1) two genes for detoxifying enzymes, NAD(P)H quinone oxidoreductase (NQO1) and glutathione S-transferases and (2) antioxidant genes such as heme oxygenase 1 and γ-glutamylcysteine synthase [63] (Figure 3). With continuous regulation in vivo, after redox levels reach equilibrium, Keap1 translocates into the nucleus to escort Nrf2 out of the nucleus for proteasomal degradation in the cytoplasm, thus terminating Nrf2 activity [64].

#### 4.1.3. Keap1/Nrf2/ARE and Oxidative Stress

Keap1 is activated during oxidative stress and has a huge role in bone remodeling and inhibition of apoptosis. Nrf2, a regulator of bone healing, was found to play an important role in bone remodeling [65]. Nrf2 is activated during the remodeling of healthy bone and fracture healing. In Nrf2-deficient mice, tissues for cartilage healing suffer high levels of oxidative stress damage, resulting in reduced strength and stability of the fractured bones and impeded bone formation and healing. In cardiomyocytes exposed to hypoxic conditions, oxidative stress is induced in the cells with a concomitant increase in the forkhead box protein O6 (FOXO6) expression. Downregulation of FOXO6 expression by gene knockdown promotes sirtuin6 (SIRT6) expression, which ultimately enhances Nrf2-mediated activation of antioxidant signaling and inhibits apoptosis [66]. In oxidative stress experiments in the jejunum of weaned heifers fed zearalenone (ZEA), there was a linear and quadratic decrease in the relative expression of total SOD and GSH-Px activities and reduced levels of Keap1 mRNA and protein in the jejunum (*p* < 0.05) with increasing dietary ZEA concentrations, while the levels of Nrf2, GPx1, NQO1, and glutamate-cysteine ligase mRNA and relative expression of modifier subunits of protein increased linearly and quadratically (*p* < 0.05), suggesting that ZEA-induced jejunal oxidative stress promotes the expression of downstream antioxidant target genes expressing NQO1, HO1, and glutamate-cysteine ligase against oxidative stress by upregulating the Keap1/Nrf2 signaling [67].

### 4.2. MAPK Signaling Pathway

The MAPK signaling pathway is an important oxidative stress-sensitive transduction pathway that plays an important role in oxidative stress. In-depth studies on the MAPK signaling pathway in skeletal muscle are an important aspect in resolving muscle damage under stress conditions. It is a well-studied classical signaling pathway with multiple important functions in regulating cell growth, metabolism, mutagenesis, transcription, translation, and recombination [68]. The most studied MAPK-regulated cellular functions include extracellular signal-regulated kinase (ERK), c-Jun N-terminal kinase, and p38 [69] (Figure 4).

The MAPK family is an important component of the oxidative stress-sensitive signaling pathway that is involved in regulating cellular responses such as growth, development, differentiation, and apoptosis when cells are exposed to external stimuli (e.g., growth factors, cytokines, neurotransmitters, and hormones) [70,71,72,73]. Research on the relationship between stress and MAPK signaling pathway is a topic of interest. It is evident that activation of the p38MAPK signaling pathway is an important manifestation of organismal oxidative stress. Shin et al. [74] found that overexpression of p38MAPK inhibited apoptosis induced by oxidative stress and played a protective role for cells. Their study confirmed that the MAPK signaling pathway is activated in oxidative stress and is involved in mediating oxidative stress-induced cellular damage [75,76,77].

The MAPK signaling pathway is significant in the regulation of skeletal muscle development. Will et al. (2013) found that leptin could affect the growth of porcine skeletal muscle myogenic cells by altering the expression of key genes in the MAPK signaling pathway [78]. This was also corroborated by recent studies that the MAPK signaling pathway plays a key role in regulating skeletal muscle growth and development during oxidative stress [79,80,81]. MAPK may be involved in regulating genes controlled by the expression of the NF-ĸB signaling pathway (such as several antioxidant enzyme-induced NF-ĸB signaling pathways). ERK1/2 phosphorylates transcription factors associated with growth and development [82]. Moreover, several important adaptations in skeletal muscle, such as mitosis, organ hypertrophy, and myofiber conversion, are regulated by the MAPK signaling pathway, and these adaptations play an important role in determining oxidative–antioxidative homeostasis in the intracellular environment.

### 4.3. AMPK Signaling Pathway

AMPK is a highly conserved regulator of cellular energy metabolism [83,84] that is activated by multiple upstream signals, such as cellular AMP:ATP and ADP:ATP ratios, CaMKKb, and LKB1. AMPK participates in the regulation of cellular events, such as the assembly of the tight junction, cell proliferation, and differentiation [85,86,87,88,89]. Inhibition of the AMPK pathway has a suppressive effect on barrier function [90], but activation of the pathway can facilitate the assembly of tight junctions [91]. The study of Yang et al. (2018 a, b) indicated that heat stress induced the dephosphorylation of AMPK in Sertoli cells, but this was contrary to its effect in other somatic cells [92]. Furthermore, the findings indicated that in cultured immature boar Sertoli cells, heat stress inhibited the AMPK signaling pathway by suppressing the expression of CaMKKb, and activation or overexpression of AMPK could reverse the heat-induced downregulation of tight junction proteins [93].

Numerous studies showed that accumulation of ROS could lead to the dysfunction of tight junction via various signaling pathways [94,95]. In the gill of fish, ROS induced tight junction damage through Nrf2, mTOR, and NF-κB signaling molecules [96]. LPS-induced excessive generation of ROS led to disruption of gut epithelial barrier in vitro [97]. Activation of AMPK also improved LPS-induced dysfunction of the blood–brain barrier via suppressing the generation of ROS in mice and human brain [98,99]. To investigate whether heat stress-induced oxidative stress participates in AMPK-mediated changes in the tight junction, Yang et al. (2020) used NAC to inhibit the overproduction of ROS [100]. Pre-treatment with NAC decreased the level of ROS and increased the expression of CaMKKb and the phosphorylation level of AMPK, and this, in turn, reversed the heat-induced downregulation of tight junction proteins in immature boar Sertoli cells. Thus, these findings implied that heat stress-induced oxidative stress mediated changes in tight junction proteins via the CaMKKb-AMPK axis.

Several studies have reported that AMPK played a central protective role in attenuating oxidative injury and regulating mitochondrial function [101,102]. AMPK has been implicated in being involved in the regulation of oxidative stress, through phosphorylating some transcription factors, including the master transcriptional regulator of lysosomal genes, TFEB. TFEB is tightly connected with stress, with non-stressed conditions reducing hyperphosphorylation and cytoplasmic reservation, and stress conditions facilitating hypophosphorylation and nuclear translocation. It was demonstrated that addition with curcumin in IPEC-J2 cells challenged with H_2_O_2_ unregulated the AMPK phosphorylation and TFEB level in the nucleus [103].

## 5. Nutritional Modulation Measures to Mitigate Oxidative Stress

Oxidative stress is common in the intensive production of pigs, and it poses a greater risk to animal health and increases feeding costs. Therefore, it is important to develop effective oxidative stress mitigation techniques.

Under oxidative stress, body nutrient metabolism direction changes, additional supplementation of certain nutrients or non-nutritional additives help reduce stress. Commonly used supplements belong to the following categories: 1. amino acid, such as cysteine, arginine, and tryptophan; 2. Vitamins, such as vitamins E, A, and C; 3. mineral elements, such as zinc, copper, manganese, selenium, etc.; 4. natural compounds, such as curcumin and resveratrol.

### 5.1. Functional Amino Acids

*Cysteine*: In a piglet model, cysteine supplementation significantly increased intestinal GSH content, activities of antioxidant enzymes SOD, GSH-Px, and catalase, and further alleviated inflammatory response and intestinal damage induced by LPS [104]. The supplementation of 1.2 g/kg cysteine precursor (acetylcysteine) in development-delayed piglets upregulated the expression of GSH biosynthesis related genes in liver and increased the GSH content, which significantly improved the growth and development of piglets [105]. In the lipopolysaccharide model, dietary acetylcysteine supplementation also showed significant anti-inflammatory effects on lipopolysaccharide-induced piglet inflammation oxidative stress [106,107].

*Arginine*: Arginine can indirectly regulate the oxidation system in the body through a variety of ways. In the piglet model of oxidative stress, arginine significantly increased the activities of GSH-Px and SOD, and alleviated the oxidative damage induced by diquat [108]. The experiment of finishing pigs showed that arginine improved the antioxidant capacity of muscle and meat quality indexes of pork [109]. Recent study found that L-arginine inhibited inflammatory response and oxidative stress induced by lipopolysaccharide via arginase-1 signaling in IPEC-J2 cells [110].

*Tryptophan*: Tryptophan and its metabolites have also been reported to have antioxidant properties [111]. Studies have shown that oxidative stress induced by diquat significantly reduced blood tryptophan content in piglets [112], while feeding tryptophan in suckling piglets inhibited the secretion of stress hormone, and improved production performance [113]. In vitro studies showed that tryptophan could activate the mTOR signaling pathway in intestinal epithelial cells of piglets, and upregulated the expression of amino acid transporters and tight junction proteins, which had a positive effect on intestinal function [114]. However, the addition of high dose tryptophan had a certain degree of toxicity on the intestinal morphology of piglets [115]. Melatonin is one of the main metabolites of tryptophan. The study of Ji et al. [116] suggested that melatonin treatment decreased PM2.5-induced oxidative stress level in the brains and lungs and relieved airway inflammation and chronic cough. In vitro experiments found that melatonin alleviated oxidative stress injury induced by H_2_O_2_ in pigs and promoted the development of pig embryos [117].

### 5.2. Vitamin-Based Antioxidant Supplementation

*Vitamin E:* Vitamin E is the most important fat-soluble antioxidant [118] and benefits the organism by reducing oxidative stress and affecting cytokine expression [119]. Vitamin E was found to resist PVC-induced sperm damage and embryotoxicity, increase sperm count and viability in animals, improve sperm DNA damage, increase in vitro fertilization rates, and promote embryo development [120]. In another study, piglets fed with peroxidized soybean oil exhibited oxidative stress and adversely affected growth, and supplementation with vitamin E increased serum vitamin concentrations (*p* < 0.001) compared to control and polyphenol treatment (1.98, 1.25, and 1.26 mg/kg) [121]. In that study, both vitamin E and polyphenols improved antioxidant capacity in piglets, although there was no significant change in growth performance or oxidative status compared to the control group.

*Vitamin A and beta-carotene:* Structurally, β-carotene consists of two vitamin A molecules and is, thus, used as a source of vitamin A [122]. Based on the needs of the body, the negative feedback regulation mechanism causes the conversion of β-carotene to vitamin A, and this mechanism effectively intercepts the problem of excessive accumulation of vitamin A in the body [123]. Deficiency of vitamin A in cells results in ROS, mitochondrial dysfunction, PARP-1-dependent energy deprivation, and programmed T-cell death, which decreases the immunity of the organism [124]. Ahmed M. Elomda et al. [125] studied the breakdown product of β-carotene in the in vitro cultured rabbit mulberry embryos, and found that the activities of SOD and GPx antioxidant enzymes were significantly higher (*p* < 0.05) in the group with the addition of 100 or 1000 nM retinol than in the control group. Their results indicated an increased antioxidant capacity, in addition to a significant upregulation of key genes for embryonic development (e.g., GJA1 and POU5F1 genes, gap junction protein α1, and POU 5-like homology cassette 1), which cause significantly enhanced rabbit embryo development (*p* < 0.001). Vitamin A supplementation (129 mg/kg) in lactating and post-weaning rats attenuated oxidative stress and metabolic disturbances due to a high-fat diet, significantly reducing the body weight and amount of the white adipose tissue in rat pups, but increased the amount of brown adipose tissue. This provides one of the possible reasons for obesity/overweight in early life.

*Vitamin C:* Vitamin C protects cells from damage by acting as an antioxidant to scavenge ROS and prevent lipid peroxidation and protein alkylation through the vitamin E-dependent pathway [126]. Dietary supplementation with vitamin C at any dose (LVc and HVc) significantly improved growth and feed utilization in fish and increased animal body weight [127]. The role of vitamin C in metabolic syndrome was reviewed in 2020 [128], and positive outcomes of vitamin C were found to be mediated in part through its antioxidant and anti-inflammatory properties. Vitamin C supplementation significantly increased SOD and GPx levels and reduced MDA levels compared to those in controls (*p* < 0.05). Reduced oxidative stress induced by the insecticide mixture fipronil (Fip) + pirimiphos-methyl (Pyr), reduced insecticide neurotoxicity, and improved motility in juvenile zebrafish, possibly by mechanisms that alter the expression of hypothalamic–pituitary–gonadal axis genes [129] and increase cortisol levels [130].

### 5.3. The Mineral Elements

*Copper:* One of the enzymes that neutralize oxidative stress is Cu/Zn SOD, which has some ability to scavenge ROS; hence, adequate levels of copper in the body are important to mitigate oxidative stress [131,132]. The addition of cholesterol to the animal diet can lead to copper deficiency and, thus, promote oxidative damage [133]. Copper deficiency can also alter the microbiome of animals [134]. Supplementation of weaned piglets’ diets with copper nanoparticles enhanced their antioxidant capacity, clearly indicating that the addition of 100 mg/kg body weight of copper nanoparticles to the diet may be a potential alternative for weaned piglets [135].

*Zinc:* Zinc is one of the essential trace elements that are present in many metabolic enzymes, transcription factors, and cellular signaling proteins and acts as a catalyst for enzymes in DNA replication, gene transcription, and RNA and protein synthesis [136,137]. A possible mechanism for the antioxidant properties of zinc is Cu-Zn-SOD, which is an important cellular resistance component to the first line of defense against ROS [138]. The addition of ZnAA (zinc-amino acid complex) could alleviate oxidative stress by reducing plasma MDA levels and decreasing GPx activity [139]. Increased intestinal villi length and the ratio of villi length to crypt depth indicated improved intestinal morphology, increased feed utilization efficiency, and improved body weight and FCR. In another study [140], the addition of 200 mg/kg Zn could effectively protect weaned piglets from oxidative stress by improving the antioxidant system, and different Zn sources (2-hydroxy-4 methyl-thio butanoic acid (HMZn) vs. ZnSO_4_) did not affect growth performance during the first two weeks. 2-hydroxy-4 methyl-thio butanoic acid HMZn increased serum SOD and GPx activities and total antioxidant capacity (T-AOC) (*p* < 0.05). Compared to the diquat group, different sources of zinc increased serum GPx and T-AOC activities and increased relative liver and kidney Nrf2, SOD1, and GPx mRNA expression (*p* < 0.05). The addition of HMZn increased the depth of jejunal villi (*p* > 0.05), and downregulated mRNA expression of inflammatory factors in the small intestine, liver, and kidney, ultimately indicating that HMZn is a more effective source of zinc.

*Selenium:* The antioxidant effect of selenium is mainly reflected in its role as a component of GSH-Px, which is an important antioxidant enzyme in animals that protects the cell membranes of the animal organism from oxidative damage. Selenium deficiency generates oxidative stress through the regulation of selenoproteins, and subsequent activation of inflammatory responses via NF-κB, HIF-1α, and NLRP3 pathways, ultimately leading to organ damage [141,142,143]. Doan et al. (2020) studied the addition of different selenium sources, such as sodium selenite, soy protein-chelated Se, and selenized yeast [144]. To study the oxidative stress induced by diquat, they categorized the nursery pigs into five groups, (1) the negative control (NC): fed with basal diet and injected with sterile saline via the intraperitoneal cavity, (2) the positive control (PC): fed with basal diet and injected with diquat solution, (3) fed with a basal diet supplemented with 0.3 mg/kg of Se from sodium selenite (SS), (4) soy protein-chelated Se (SC), and (5) selenized yeast (SY). Organic selenium-enhanced endogenous antioxidant activity in all aspects compared to the PC group (*p* < 0.05), with SY being the most effective, ultimately indicating that the addition of a selenium-enriched yeast source at a dose of 0.3 mg/kg selenium was the most effective. In the mouse model of intestinal oxidative stress, a new form of selenium nanoparticle (biogenic nanoselenium (BNS) particles) were found to protect the mouse intestinal barrier function and preserve intestinal redox homeostasis more efficiently than Se-Met and Nano-Se [145]. In vitro experiments with porcine jejunum epithelial (IPEC-J2) cells verified the stronger epithelial barrier-protecting effect of BNS particles against oxidative stress, with reduced cell apoptosis and an improved cell redox state. Furthermore, the study of Sun et al. (2020) reported that selenium supplementation protects against oxidative stress-induced cardiomyocyte cell cycle arrest through activation of PI3K/Akt [146].

### 5.4. The Natural Compounds

*Curcumin:* Curcumin is a compound from turmeric that has certain anti-inflammatory, antioxidant, antiproliferative, and antiangiogenic activities [147,148]. Curcumin can scavenge free radicals and carry out antioxidant functions by inducing the Nrf2 signaling pathway [149]. The addition of 400 mg/kg curcumin to the basal diet was effective in alleviating intrauterine growth retardation (IUGR)-induced intestinal oxidative stress in piglets and improved intestinal antioxidant function [150]. The work of Cao et al. (2020) proposed that curcumin could effectively ameliorated oxidative stress, enhanced intestinal barrier function and mitochondrial function through induction of Parkin-dependent mitophagy via AMPK activation and subsequent TFEB nuclear translocation [103].

*Resveratrol:* Resveratrol is a compound derived from grapes and wine [151] that reduces inflammation and regulates redox mechanism through Sir2-related enzymes, the Sirtuins1 (SIRT1)/peroxisome proliferator-activated receptor gamma coactivator 1α (PGC-1α) axis, and signaling pathways such as PI3K/Akt/mTOR [152,153]. Dietary supplementation of 300 mg/kg resveratrol fed to pregnant and lactating sows increased the antioxidant capacity of the sow’s placenta and milk compared to the control group [154], which also increased the antioxidant level of piglets as well as the litter weaning and piglet weaning weights (*p* < 0.05). These data from Cao et al. (2019) indicated that resveratrol was effective in protecting the intestinal barrier, improving the redox status, alleviating mitochondrial damage and inducing mitophagy in piglets challenged with diquat [155]. Table 1 summarizes the antioxidant effects of some natural compounds.

## 6. Conclusions 

Therefore, enhancing the antioxidant capacity of pigs by supplementing/adding relevant nutrients is a good practice to enhance the amount of enzymatic/non-enzymatic antioxidants or stimulate the expression of antioxidant genes, as well as to control the key sites of the redox signaling pathway in animal production.

At present, there are still many topics that need to be studied urgently in the field of pig oxidative stress and nutrition regulation. (1) Pig weaning is a comprehensive process, and the specific mechanism of weaning oxidative stress and its accompanying or leading role in the process of weaning stress still need to be further studied. (2) Oxidative stress is closely related to a variety of human metabolic diseases, and pigs are the most ideal model animal for human disease research [81]; however, there are still few studies on pig oxidative stress and related human diseases. (3) There is a lack of mature and stable models of pig oxidative stress at home and abroad, which directly restricts the study of pig oxidative stress. Methionine overdose can induce oxidative stress. Thus, it is unknown whether oxidative stress is induced by increasing dietary methionine content in a pig model. (4) The research on oxidative stress of pigs at home and abroad is mainly focused on the piglet stage. There is no systematic comparison of the oxidative stress response in different growth stages of pigs, and different oxidative stress models have different targeting organs for oxidative damage in pigs, which hinders the systematic study of the mechanism of oxidative stress in pigs.

## Figures and Tables

**Figure 1 animals-11-01384-f001:**
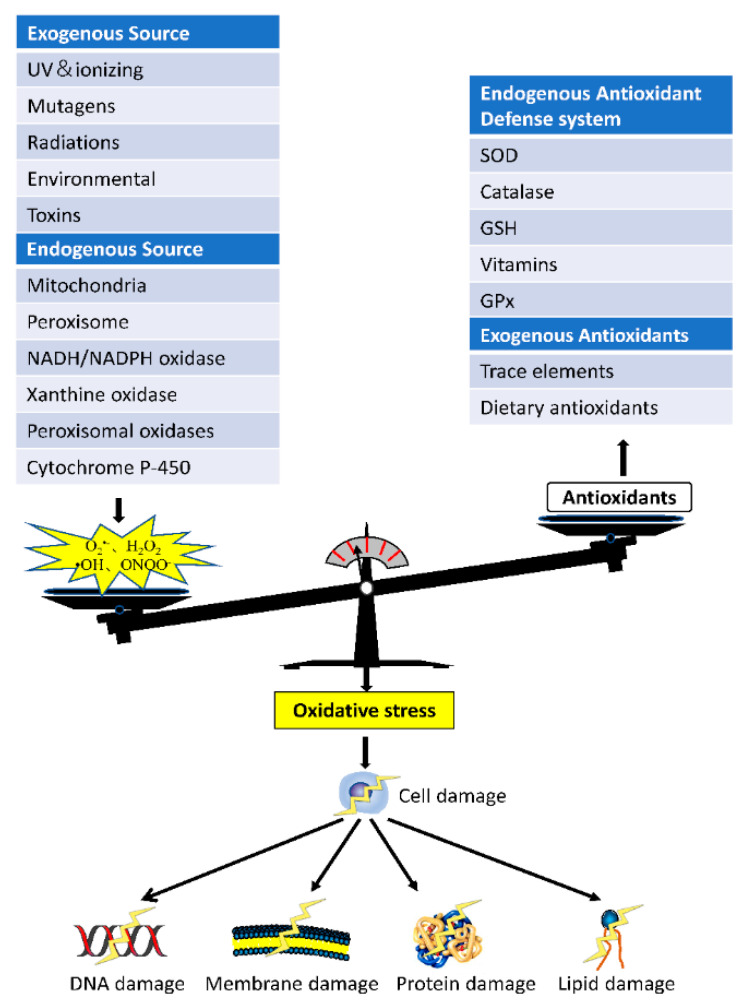
When the oxidative system in the body is stronger than the antioxidant system, the generation of excess ROS cannot be scavenged in time. This disrupts the homeostasis of redox balance in the body and causes oxidative stress, ultimately leading to DNA, membrane, protein, and lipid damage.

**Figure 2 animals-11-01384-f002:**
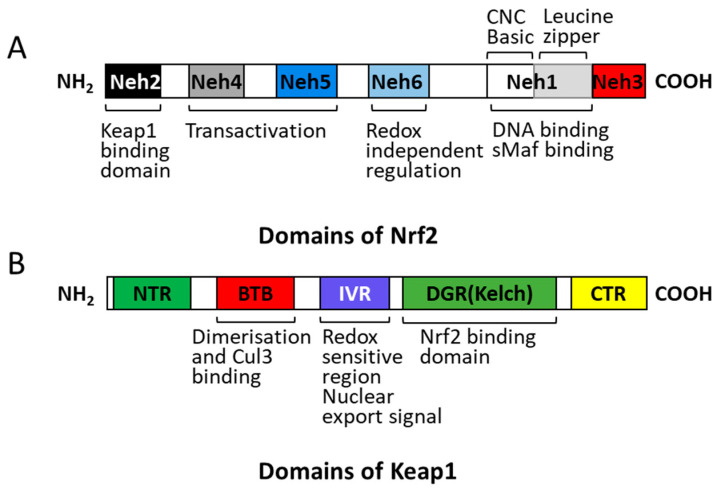
The structural regions of Nrf2 and Keap1 proteins. A, Domains of Nrf2; B, Domains of Keap1.

**Figure 3 animals-11-01384-f003:**
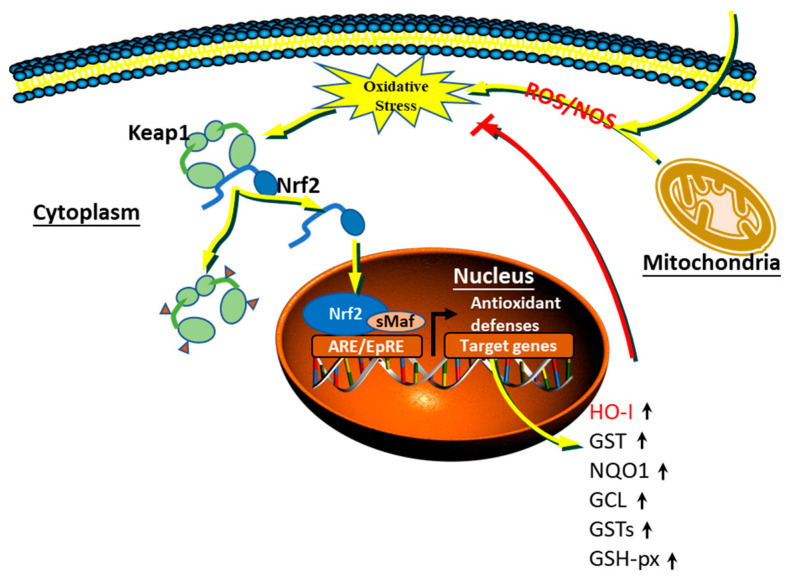
Signaling pathway of Keap1/Nrf2/ARE under oxidative stress.

**Figure 4 animals-11-01384-f004:**
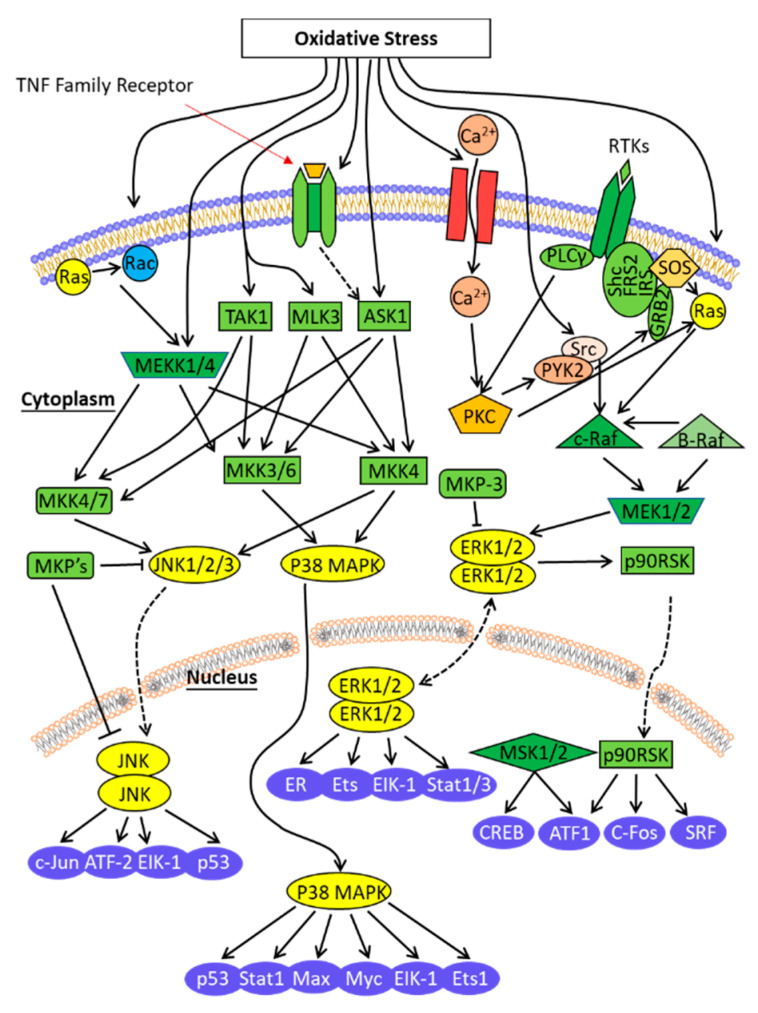
Signaling pathway of MAPK under oxidative stress.

**Table 1 animals-11-01384-t001:** The antioxidant effects of some natural compounds.

Active Ingredients	Experimental Model	Doses	Efficacy	Ref
Resveratrol	diquat-induced intestinal barrier function in piglets.	100 mg/kg, 14-day	protecting the intestinal barrier antioxidant capacity↑ alleviating mitochondrial damage	[155]
Curcumin	Intrauterine growth retardation piglets	400 mg/kg, 24-day	Growth performance↑ hepatic antioxidant capacity↑ Nrf2 and Hmox1 levels↑	[156]
Quercetin	transport-induced intestinal injury pigs	25 mg/kg, 4-week	alleviates intestinal injury during transport by modulation of intestinal oxidative status and inflammation	[157]
Proanthocyanidin	weaned piglet	250 mg/kg, 28-day	Resisting intestinal oxidative stress by increasing diversity and improving the balance of gut microbes. Piglets had better growth performance and reduced diarrhea incidence.	[158]
Garcinol	finishing pigs	600 mg/kg	antioxidant capacity↑ growth performance ↑ pork quality↑	[159]
Protocatechuic acid	weaned piglet model challenged with lipopolysaccharide (LPS)	4000 mg/kg	Protective effects on oxidative stress, inflammation, and intestinal barrier function through regulation of intestinal flora	[160]
Artemisia annua L.	heat-stressed sows	1.0 g/kg	antioxidant capacity↑ increased piglet weaning weight and the activities of T-SOD and T-AOC in serum and promote the intestinal barrier integrity.	[161]
Icariin	enterotoxigenic Escherichia coli-induced intestinal epithelial barrier disruption in piglets	1 g/kg BW	expression of p38 MAPK↑ antioxidant capacity↑	[162]
Dioscin	Chinese miniature pigs (male, 20–30 kg, 1–2 month)	80 mg/kg	regulating oxidative stress and inflammation via Sirt1/Nrf2 and p38 MAPK pathways	[92]
Chitosan (CS)	diquat-induced oxidative stress in weaned piglets	500 mg/kg	antioxidant capacity↑ anti-inflammatory↑	[163]
Grape pomace	weaned piglets	9% GP	antioxidant capacity↑ growth performance↑ pork quality↑	[164]
Soybean isoflavone (ISF)	young piglets fed oxidized fish oil	20 mg/kg	intestinal morphology↑ antioxidant capacity↑ immune function↑	[165]
Konjac flour (KF)	Gestating sows	2.2%	intestinal morphology↑ antioxidant capacity↑ insulin sensitivity	[166]
Polyphenolic (byproduct from olive mill wastewater processing)	piglets		total antioxidant capacity in plasma and tissues↑ CARB and TBARS in plasma and tissues↓	[167]
Oregano essential oil (OEO)	large white sows	15 mg/kg	antioxidant capacity↑ Performance of future generations↑	[168]
Red ginseng (Panax ginseng)	Isoproterenol-Induced myocardial infarction in Porcine	(250 and 500 mg/kg; gastric gavages, respectively) for 9 days	antioxidant capacity↑ significant cardioprotective potential, adjunct in the treatment and prophylaxis of myocardial infarction.	[169]
Shenyuan	A porcine model of acute myocardial infarction (AMI)	400 mg/kg·d	antioxidant capacity↑ cardiomyocyte apoptosis↓ therapeutic role in improving the natural process of AMI	[170]
Verbascoside	piglets	5 mg/kg	stress biomarkers in swine gut↓	[171]

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
