# Peer review of "Research Progress on Oxidative Stress and Its Nutritional Regulation Strategies in Pigs"

_animals, 2021, doi:10.3390/ani11051384_

Round 1

Reviewer 1 Report

Minor comments:

In the present study, the authors reviewed the latest advances in the effects of oxidative stress on animals and their nutritional regulation, and analyzed the regulatory effects of nutritional additives on oxidative stress in order to provide some ideas for reducing the harm caused by oxidative stress in animal production. The analysis in this paper is comprehensive and cutting-edge, and it is clearly organized, which has important reference significance for reducing the harm caused by oxidative stress in animal production. However, some minor deficiencies must be corrected before consideration for publication.

  1. English style in the manuscript should be improved with help of native skilled English speaker for vantage look. I have listed only a few examples here, so please check the full text carefully.

Line 75: change “Damage to the animal by oxidative stress” to “Damage to animals by oxidative stress”

Line 76: change “Effects of oxidative stress on various organs of the animal” to “Effects of oxidative stress on animal organs”

Lines 77-80: change “In the state of oxidative stress, accumulated ROS, if not cleared in time, damage the body's DNA and proteins through a series of pathways, and these damages promote cellular senescence, apoptosis, and death, which eventually amplifies into organismal damage.” to “Under the state of oxidative stress, if the accumulated ROS is not cleared in time, it will damage the body's DNA and proteins through a series of pathways, and these damages will promote cell senescence, apoptosis and death, and eventually expand into organismal damage.”

Line 89: change “Cardiomyocyte oxidative stress” to “Oxidative stress of cardiomyocytes”

Lines 91-92: change “Tebuconazole-induced oxidative stress in the heart causes injurious effects on the myocardium characterized by inflammatory……” to “The myocardial damage caused by tebuconazole-induced oxidative stress was characterized by inflammatory cell infiltration…..”

Lines 194-196: Please check this sentence

Author Response

We appreciate the valuable suggestion. The manuscript has been rewritten. And the mentioned parts have been deleted.

Reviewer 2 Report

In the last decade, the topic of oxidative stress in animal productions has been covered by many review papers and therefore this manuscript aims to give a new viewpoint in respect to the previous papers. However, this manuscript is very ambitious because it considers all terrestrial farmed animals without any distinction among mammals and avian species, monogastrics and ruminants, breeders and growing animals.

First part of the review paper involving introduction, damages to animal by oxidative stress, effects of oxidative stress on meat quality is redundant in respect to previous published review papers and lacks of effectiveness because of the too large group of considered animals. The most valuable part of the manuscript is section dedicated to oxidative stress signal pathway, because its scope is limited, and authors were able to give an overview of information available in literature. Last part dealing with nutritional modulation measures is again quite unspecific and not comprehensive for the above-mentioned reasons.

It is highly recommended to extensively revise the manuscript by limiting the scope and avoiding redundancy on well-know mechanisms extensively described in previous papers. Otherwise, this review risks to be of poor interests for Animals journal readers.

Author Response

According to the suggestions, we focused on pigs, and supplemented some research which are about main factors in the pig production process that induce oxidative stress, model of oxidative stress in pigs and functional amino acids in alleviating oxidative stress.

Reviewer 3 Report

Dear Authors,

The paper is an excellent,  very extensive, and deep review of oxidative stress and potential nutritional strategies to prevent it. Figures are very nice and illustrative.

There just very few minor comments from the reviewer:

Line 144 not sure if M.Lv is fine or should be only Lv et al.(spelling)

Lines 236-237. Although the paper is focused on oxidative stress, since here it mentions that stress decreases pH after slaughter, and the negative consequences, perhaps it could just briefly mention the PSE meat defect due to ante-mortem stress and sudden decline of pH.

Line 281. Spelling mistake: there is a “,.” in between two sentences instead of a “.”

Line 440-459. Just for curiosity, the fact that the authors do not mention ZnO, which has been used to prevent diarrhea in weaned piglets for example, is it because of its dose-dependant negative environmental impacts? Or because the mechanism of action of this source of Zn is not associated with changes in oxidative stress pathways?

Author Response

  1. Line 281. Spelling mistake: there is a “,.” in between two sentences instead of a “.”

Response: We have revised as suggested.

  1. Line 440-459. Just for curiosity, the fact that the authors do not mention ZnO, which has been used to prevent diarrhea in weaned piglets for example, is it because of its dose-dependant negative environmental impacts? Or because the mechanism of action of this source of Zn is not associated with changes in oxidative stress pathways?

Response: We do not mention ZnO, because the mechanism of action of this source of Zn is not associated with changes in oxidative stress pathways.

Other mentioned parts have been deleted in the revision.

Reviewer 4 Report

Review

of

Recent advances in research on the effects of oxidative stress on animals and its nutritional regulation

The topic and the mode of conduct is valuable, but there are issues that are  needed to be dealt with.

Below the authors find some comments and suggestions in order to  improve the MS.

Simple summary:

’e.g., immunotherapy, environmental changes, uncomfortable temperature, feed contamination, improper transportation, and slaughter methods, etc.). These adverse stimuli eventually translate into an imbalance in redox levels in the animal, resulting in oxidative stress.’

This is a very important thought, but it is unfortunately not detailed and described in the main body of the review. Please include, because the stressors resulting oxidative stress are  a significant part of the initiation of the process.

Abstract

The abstact is clear and concise.

line 17 ’ Oxidative stress is inevitable in animal production and is one of the most important factors that affect and limit animal production.’

This is a very general statement. Is it really true for all aspects of all kinds of animal production? Please reword this sentence.

Introduction

line 66  and 69 please delete ’etc. ’

Please include a paragraph at the end of the Introduction about the main aspects this review work was planned to conduct.

line 77 What do the authors mean by ’cleared’? Detoxified, alleviated, neutralised?

Figure 1. If the intention of the authors is to list e.g the enzmyes, please make the list complete.

Please decide, whether to start a word with capital letters or not, but please do not mix them.

4.1.1. Structural domains of Keap1 and Nrf2 277

Figure 2. The structural regions of Nrf2 and Keap1 proteins

Is it an original figure, or an adapted one? Please indicate.

5.3. The natural compounds

There are several other natural (hytogenic) compounds, besides curcumin and resveratrol. please extend this list using recent relevant review and experimental works.

4.1. Keap1/Nrf2/ARE signaling pathway 

If this signalling pathway is strengthened with a figure, please also include one about the MAPK signalling pathway as well.

Conclusions

It contains mostly general sentences about this topic. What are the key findings of this review? What novelty does it have compared to previous review works?

Please, include these thoughts into the conclusions, and if possible indicate future implications based on these findings.

Formatting issues:

line 95 heart[14]

line 109 disease[20],

line 126 depth[26].

Author Response

  1. This is a very important thought, but it is unfortunately not detailed and described in the main body of the review. Please include, because the stressors resulting oxidative stress is a significant part of the initiation of the process.

Response: We appreciate the valuable suggestion. We have supplemented some research which are about main factors in the pig production process that induce oxidative stress.

  1. line 17: Oxidative stress is inevitable in animal production and is one of the most important factors that affect and limit animal production.’

This is a very general statement. Is it really true for all aspects of all kinds of animal production? Please reword this sentence.

Response: We appreciate the valuable suggestion. We have rewritten the Abstract.

  1. Please include a paragraph at the end of the Introduction about the main aspects this review work was planned to conduct.

Response: We appreciate the valuable suggestion. We have supplemented a paragraph as suggestion.

  1. Figure 1. If the intention of the authors is to list e.g the enzmyes, please make the list complete.

Please decide, whether to start a word with capital letters or not, but please do not mix them.

Response: We appreciate the valuable suggestion. We have deleted the “Enzymes” and revised the first words with capital letters.

  1. 4.1.1. Structural domains of Keap1 and Nrf2 277

Figure 2. The structural regions of Nrf2 and Keap1 proteins

Is it an original figure, or an adapted one? Please indicate.

Response: It’s an adapted figure.

  1. 5.3. The natural compounds

There are several other natural (hytogenic) compounds, besides curcumin and resveratrol. please extend this list using recent relevant review and experimental works.

Response: We appreciate the valuable suggestion. We have supplemented the antioxidant effects of some other natural compounds in a table.

  1. 4.1. Keap1/Nrf2/ARE signaling pathway 

If this signalling pathway is strengthened with a figure, please also include one about the MAPK signalling pathway as well.

Response: We appreciate the valuable suggestion. We have supplemented a figure about the MAPK signalling pathway named Figure 4.

  1. Conclusions

It contains mostly general sentences about this topic. What are the key findings of this review? What novelty does it have compared to previous review works?

Please, include these thoughts into the conclusions, and if possible indicate future implications based on these findings.

Response: We appreciate the valuable suggestion. We have rewritten the conclusions as suggested.

Round 2

Reviewer 2 Report

I appreciate the big efforts of the authors in revising manuscript following the suggestions made by all reviewers.